# "Hey, that's not an ODE": Faster ODE Adjoints with 12 Lines of Code

## Abstract

Neural differential equations may be trained by backpropagating gradients via the adjoint method, which is another differential equation typically solved using an adaptive-step-size numerical differential equation solver. A proposed step is accepted if its error, *relative to some norm*, is sufficiently small; else it is rejected, the step is shrunk, and the process is repeated. Here, we demonstrate that the particular structure of the adjoint equations makes the usual choices of norm (such as $L^2$) unnecessarily stringent. By replacing it with a more appropriate (semi)norm, fewer steps are unnecessarily rejected and the backpropagation is made faster. This requires only minor code modifications. Experiments on a wide range of tasks—including time series, generative modeling, and physical control—demonstrate a median improvement of 40% fewer function evaluations. On some problems we see as much as 62% fewer function evaluations, so that the overall training time is roughly halved.

## 1 Introduction

We begin by recalling the usual set-up for neural differential equations.

### 1.1 Neural ordinary differential equations

The general approach of neural ordinary differential equations (E, 2017; Chen et al., 2018) is to use ODEs as a learnable component of a differentiable framework. Typically the goal is to approximate a map $x \mapsto y$ by learning functions $\ell_1(\,\cdot\,, \phi)$, $\ell_2(\,\cdot\,, \psi)$ and $f(\,\cdot\,, \,\cdot\,, \theta)$, which are composed such that

$$z(\tau) = \ell_1(x, \phi), \qquad z(t) = z(\tau) + \int_\tau^t f(s, z(s), \theta)\,\mathrm{d}s \quad \text{and} \quad y \approx \ell_2(z(T), \psi). \tag{1}$$

The variables $\phi$, $\theta$, $\psi$ denote learnable parameters and the ODE is solved over the interval $[\tau, T]$. We include the (often linear) maps $\ell_1(\,\cdot\,, \phi)$, $\ell_2(\,\cdot\,, \psi)$ for generality, as in many contexts they are important for the expressiveness of the model (Dupont et al., 2019; Zhang et al., 2020), though our contributions will be focused around the ODE component and will not depend on these maps.

Here we will consider neural differential equations that may be interpreted as a neural ODE.

### 1.2 Applications

Neural differential equations have to the best our knowledge three main applications:

1. Time series modeling. Rubanova et al. (2019) interleave Neural ODEs with RNNs to produce ODEs with jumps. Kidger et al. (2020) take $f(t, z, \theta) = g(z, \theta)\frac{\mathrm{d}X}{\mathrm{d}t}(t)$, dependent on some time-varying input $X$, to produce a neural *controlled* differential equation.

2. Continuous Normalising Flows as in Chen et al. (2018); Grathwohl et al. (2019), in which the overall model acts as coupling or transformation between probability distributions,

3. Modeling or controlling physical environments, for which a differential equation based model may be explicitly desired, see for example Zhong et al. (2020).

### 1.3 ADJOINT EQUATIONS

The integral in equation (1) may be backpropagated through either by backpropagating through the internal operations of a numerical solver, or by solving the backwards-in-time *adjoint equations* with respect to some (scalar) loss $L$.

$$a_z(T) = \frac{\mathrm{d}L}{\mathrm{d}z(T)}, \qquad a_z(t) = a_z(T) - \int_T^t a_z(s) \cdot \frac{\partial f}{\partial z}(s, z(s), \theta)\,\mathrm{d}s \qquad \text{and} \qquad \frac{\mathrm{d}L}{\mathrm{d}z(\tau)} = a_z(\tau),$$

$$a_\theta(T) = 0, \qquad a_\theta(t) = a_\theta(T) - \int_T^t a_z(s) \cdot \frac{\partial f}{\partial \theta}(s, z(s), \theta)\,\mathrm{d}s \qquad \text{and} \qquad \frac{\mathrm{d}L}{\mathrm{d}\theta} = a_\theta(\tau),$$

$$a_t(T) = \frac{\mathrm{d}L}{\mathrm{d}T}, \qquad a_t(t) = a_t(T) - \int_T^t a_z(s) \cdot \frac{\partial f}{\partial s}(s, z(s), \theta)\,\mathrm{d}s \qquad \text{and} \qquad \frac{\mathrm{d}L}{\mathrm{d}\tau} = a_t(\tau),$$

$$\tag{2}$$

These equations are typically solved together as a joint system $a(t) = [a_z(t), a_\theta(t), a_t(t)]$. (They are already coupled; the latter two equations depend on $a_z$.) As additionally their integrands require $z(s)$, and as the results of the forward computation of equation (1) are usually not stored, then the adjoint equations are typically additionally augmented by recovering $z$ by solving backwards-in-time

$$z(t) = z(T) + \int_T^t f(s, z(s), \theta)\mathrm{d}s. \tag{3}$$

### 1.4 CONTRIBUTIONS

We demonstrate that the particular structure of the adjoint equations implies that numerical equation solvers will typically take too many steps, that are too small, wasting time during backpropagation. Specifically, the accept/reject step of adaptive-step-size solvers is too stringent.

By applying a correction to account for this, we demonstrate that the number of steps needed to solve the adjoint equations may be reduced by typically about 40%. We observe improvements on some problems by as much as 62%. Factoring in the forward pass (which is unchanged), the overall training time is roughly halved. Our method is hyperparameter-free and requires no tuning.

We do not observe any change in model performance, and at least with the `torchdiffeq` package (our chosen differential equation package), this correction may be applied with only 12 lines of code.

## 2 METHOD

### 2.1 NUMERICAL SOLVERS

Both the forward pass given by equation (1), and the backward pass given by equations (2) and (3), are solved by invoking a numerical differential equation solver. Our interest here is in adaptive-step-size solvers. Indeed the default choice for solving many equations is the adaptive-step-size Runge–Kutta 5(4) scheme of Dormand–Prince (Dormand & Prince, 1980), for example as implemented by `dopri5` in the `torchdiffeq` package or `ode45` in MATLAB.

A full discussion of the internal operations of these solvers is beyond our scope here; the part of interest to us is the accept/reject scheme. Consider the case of solving the general ODE

$$y(t) = y(\tau) + \int_\tau^t f(s, y(s))\,\mathrm{d}s,$$

with $y(t) \in \mathbb{R}^d$.

Suppose for some fixed $t$ the solver has computed some estimate $\widehat{y}(t) \approx y(t)$, and it now seeks to take a step $\Delta > 0$ to compute $\widehat{y}(t + \Delta) \approx y(t + \Delta)$. A step is made, and some candidate $\widehat{y}_{\mathrm{candidate}}(t + \Delta)$ is generated. The solver additionally produces $y_{\mathrm{err}} \in \mathbb{R}^d$ representing an estimate of the numerical error made in each channel during that step.

Given some prespecified absolute tolerance $ATOL$ (for example $10^{-9}$), relative tolerance $RTOL$ (for example $10^{-6}$), and (semi)norm $\|\cdot\| : \mathbb{R}^d \to [0, \infty)$ (for example $\|y\| = \sqrt{\frac{1}{d} \sum_{i=1}^{d} y_i^2}$ the RMS norm), then an estimate of the size of the equation is given by

$$SCALE = ATOL + RTOL \cdot \max(\widehat{y}(t), \widehat{y}_{\text{candidate}}(t + \Delta)) \in \mathbb{R}^d, \tag{4}$$

where the maximum is taken channel-wise, and the error ratio

$$r = \left\| \frac{y_{\text{err}}}{SCALE} \right\| \in \mathbb{R} \tag{5}$$

is then computed. If $r \leq 1$ then the error is deemed acceptable, the step is accepted and we take $\widehat{y}(t+\Delta) = \widehat{y}_{\text{candidate}}(t+\Delta)$. If $r > 1$ then the error is deemed too large, the candidate $\widehat{y}_{\text{candidate}}(t+\Delta)$ is rejected, and the procedure is repeated with a smaller $\Delta$.

Note the dependence on the choice of norm $\|\cdot\|$: in particular this determines the relative importance of each channel towards the accept/reject criterion.

## 2.2 ADJOINT SEMINORMS

**Not an ODE**   The key observation is that $a_\theta$ (and in fact also $a_t$) does not appear anywhere in the vector fields of equation (2).

This means that (conditioned on knowing $z$ and $a_z$), the integral corresponding to $a_\theta$ is just an integral—not an ODE. As such, it is arguably inappropriate to solve it with an ODE solver, which makes the implicit assumption that small errors now may propagate to create large errors later.

**Accept/reject**   This is made manifest in the accept/reject step of equation (5). Typical choices of norm $\|\cdot\|$, such as $L^2$, will usually weight each channel equally. But we have just established that to solve the adjoint equations accurately, it is far more important that $z$ and $a_z$ be accurate than it is that $a_\theta$ be accurate.

**Seminorms**   Thus, when solving the adjoint equations equation (2), we propose to use a $\|\cdot\|$ that scales down the effect in those channels corresponding to $a_\theta$.

In practice, in our experiments, we scale $\|\cdot\|$ all the way down by applying zero weight to the offending channels, so that $\|\cdot\|$ is in fact a seminorm. This means that the integration steps are chosen solely for the computation of $a_z$ and $z$, and the values of $a_\theta$ are computed just by integrating with respect to those steps.

**Example**   As an explicit example, note that $a_\theta(T) = 0$. When solving the adjoint equation numerically, this means for $t$ close to $T$ that the second term in equation (4) is small. As $ATOL$ is typically also small, then $SCALE$ is additionally small, and the error ratio $r$ in equation (5) is large.

This implies that it becomes easy for the error ratio $r$ to violate $r \leq 1$, and it is easy for the step to be rejected. Now there is nothing intrinsically bad about a step being rejected—we would like to solve the ODE accurately, after all—the problem is that this is a *spurious rejection*, as the rejection occurred to ensure the accuracy of $a_\theta$, which is as already established unnecessary.

In practice, we observe that spurious rejections may occur for any $t$, not just those near $T$.

**Other channels**   In fact, essentially the same argument applies to $a_t$ as well: this does not affect the value of the vector field either. In a continuous normalising flow, the log-probability channel is also only an integral, rather than an ODE, and again the same argument may be applied.

**Does this reduce the accuracy of parameter gradients?**   One obvious concern is that we are typically ultimately interested in the parameter gradients $a_\theta$, in order to train a model; with respect to this our approach seems counter-intuitive.

However, we verify empirically that models still train without a reduction in performance. We explain this by noting that the $z$, $a_z$ channels truly are ODEs, so that small errors now do propagate to create larger errors later. Thus these are likely the dominant source of error overall.

### 2.3 CODE

Depending on the software package, the code for making this change can be trivial. For example, using PyTorch (Paszke et al., 2019) and `torchdiffeq` (Chen et al., 2018), the standard set-up requires only a few additional lines of code. The additional 12 lines are marked.

```
 1  import torchdiffeq
 2
 3  def rms_norm(tensor):                                           #
 4      return tensor.pow(2).mean().sqrt()                          #
 5                                                                  #
 6  def make_norm(state):                                           #
 7      state_size = state.numel()                                  #
 8      def norm(aug_state):                                        #
 9          y = aug_state[1:1 + state_size]                         #
10          adj_y = aug_state[1 + state_size:1 + 2 * state_size]    #
11          return max(rms_norm(y), rms_norm(adj_y))                #
12      return norm                                                 #
13                                                                  #
14  torchdiffeq.odeint_adjoint(func=..., y0=..., t=...,
15                             adjoint_options=dict(norm=make_norm(y0)))   #
```

This amounts to the extra 12 lines of code stated in the title—a number that even includes the additional whitespace and visual indents.

To keep the remainder of this discussion software-agnostic, we defer further explanation of this specific code to Appendix A.

## 3 EXPERIMENTS

We compare our proposed technique against conventionally-trained neural differential equations, across multiple tasks—time series, generative, and physics-informed. These are each drawn from the main applications of neural differential equations, discussed in Section 1.2. In every case, the differential equation solver used is the Dormand–Prince 5(4) solver "dopri5". The default norm is a mixed $L^\infty/L^2$ norm used in `torchdiffeq`; see Appendix A. For the sake of an interesting presentation, we investigate different aspects of each problem.

In each case see Appendix B for details on hyperparameters, optimisers and so on.

The code for these experiments can be found at [redacted; see supplementary material].

### 3.1 NEURAL CONTROLLED DIFFERENTIAL EQUATIONS

Consider the Neural Controlled Differential Equation (Neural CDE) model of Kidger et al. (2020).

To recap, given some (potentially irregularly sampled) time series $\mathbf{x} = ((t_0, x_0), \dots, (t_n, x_n))$, with each $t_i \in \mathbb{R}$ the timestamp of the observation $x_i \in \mathbb{R}^v$, let $X: [t_0, t_n] \to \mathbb{R}^{1+v}$ be an interpolation such that $X(t_i) = (t_i, x_i)$. For example $X$ could be a natural cubic spline.

Then take $f(t, z, \theta) = g(z, \theta)\frac{dX}{dt}(t)$ in a Neural ODE model, so that changes in $\mathbf{x}$ provoke changes in the vector field, and the model incorporates the incoming information $\mathbf{x}$. This may be thought of as a continuous-time RNN; indeed Kidger et al. (2020) use this to learn functions of (irregular) time series.

We apply a Neural CDE to the Speech Commands dataset (Warden, 2020). This is a dataset of one-second audio recordings of spoken words such as 'left', 'right' and so on. We take 34975 time series corresponding to 10 words, to produce a balanced classification problem. We preprocess the dataset by computing mel-frequency cepstrum coefficients so that each time series is then regularly spaced with length 161 and 20 channels. The data was then normalised to have zero mean and unit variance. We used the `torchcde` package (Kidger, 2020), which wraps `torchdiffeq`.

The initial map $\ell_1$ (of equation (1)) is taken to be linear on $(t_0, x_0)$. The terminal map $\ell_2$ is taken to be linear on $z(t_n)$.

Table 1: Results for Neural CDEs. The number of function evaluations (NFE) used in the adjoint equations—accumulated throughout training—are reported. Test set accuracy is additionally reported. Mean $\pm$ standard deviation over five repeats. Every model has 88,940 parameters.

| | Default norm | | Seminorm | |
|---|---|---|---|---|
| RTOL, ATOL | Accuracy (%) | Bwd. NFE ($10^6$) | Accuracy (%) | Bwd. NFE ($10^6$) |
| $10^{-3}, 10^{-6}$ | $92.6_{\pm0.4}$ | $14.36_{\pm1.09}$ | $92.5_{\pm0.5}$ | $\mathbf{8.67_{\pm1.60}}$ |
| $10^{-4}, 10^{-7}$ | $92.8_{\pm0.4}$ | $30.67_{\pm2.48}$ | $92.5_{\pm0.5}$ | $\mathbf{12.75_{\pm2.00}}$ |
| $10^{-5}, 10^{-8}$ | $92.4_{\pm0.7}$ | $77.95_{\pm4.47}$ | $92.9_{\pm0.4}$ | $\mathbf{29.39_{\pm0.80}}$ |

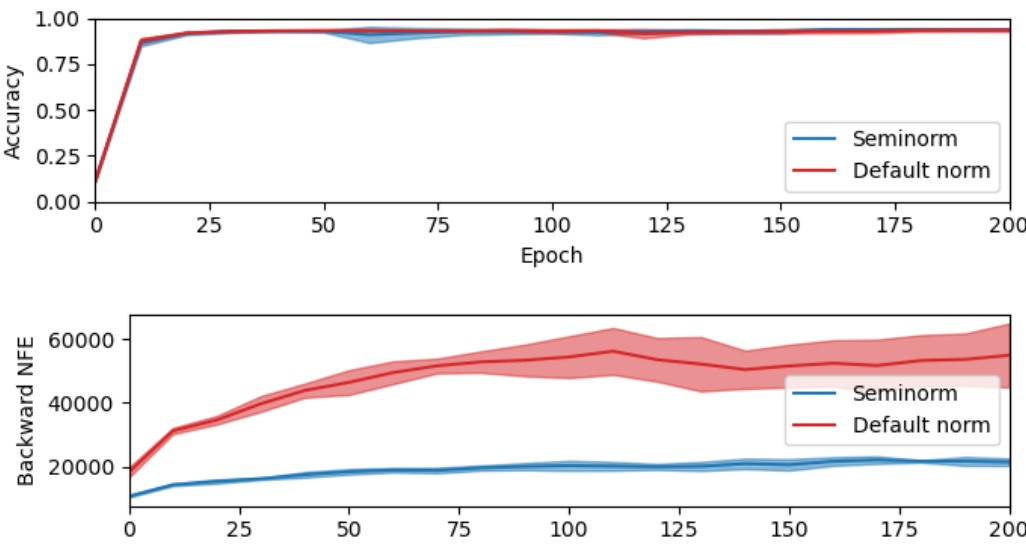

Figure 1: Mean $\pm$ standard deviation, for accuracy and the number of function evaluations (NFE) for the backward pass, during training, for Neural CDEs.

We investigate how the effect changes for varying tolerances by varying the pair $(RTOL, ATOL)$ over $(10^{-3}, 10^{-6})$, $(10^{-4}, 10^{-7})$, and $(10^{-5}, 10^{-8})$. For each such pair we run five repeated experiments.

See Table 1 for results on accuracy and number of function evaluations. We see that the accuracy of the model is unaffected by our proposed change, whilst the backward pass uses 40%–62% fewer steps, depending on tolerance.

Next, we investigate how accuracy and function evaluations change during training. See Figure 1. We see that accuracy quickly gets close to its maximum value during training, with only incremental improvements for most of the training procedure. In particular, this statement is true of both approaches, and we do not see any discrepancy between them. Additionally, we see that the number of function evaluations is much lower for the seminorm throughout training.

### 3.2 CONTINUOUS NORMALISING FLOWS

Continuous Normalising Flows (CNF) (Chen et al., 2018) are a class of generative models that define a probability distribution as the transformation of a simpler distribution by following the vector field parameterised by a Neural ODE. Let $p(z_0)$ be an arbitrary base distribution that we can efficiently sample from, and compute its density. Then let $z(t)$ be the solution of the initial value problem

$$z(0) \sim p(z_0), \qquad \frac{dz(t)}{zt} = f(t, z(t), \theta), \qquad \frac{d \log p(z(t))}{dt} = -\text{tr}\left(\frac{\partial f}{\partial z}(t, z(t), \theta)\right),$$

Table 2: Results for Continuous Normalising Flows. The number of function evaluations (NFE) used in the adjoint equations—accumulated throughout training—are reported. Test set bits/dim and number of parameters are additionally reported. Mean $\pm$ standard deviation over three repeats.

| Model | Parameters | Default norm | | Seminorm | |
|---|---|---|---|---|---|
| | | Test bits/dim | Bwd. NFE ($10^6$) | Test bits/dim | Bwd. NFE ($10^6$) |
| CIFAR-10 ($d_h = 64$) | 1,200,600 | 3.3492$\pm$0.0059 | 41.65$\pm$1.97 | 3.3428$\pm$0.0121 | **38.93$\pm$1.32** |
| MNIST ($d_h = 32$) | 281,928 | 0.9968$\pm$0.0020 | 41.50$\pm$2.35 | 0.9942$\pm$0.0013 | **37.03$\pm$0.96** |
| MNIST ($d_h = 64$) | 1,005,384 | 0.9623$\pm$0.0020 | 44.24$\pm$1.78 | 0.9603$\pm$0.0021 | **37.85$\pm$0.94** |
| MNIST ($d_h = 128$) | 3,779,400 | 0.9439$\pm$0.0030 | 48.60$\pm$2.30 | 0.9466$\pm$0.0048 | **41.84$\pm$1.92** |

for which the initial point is randomly sampled from $p(z_0)$, and for which the change in log probability density is also tracked as the sample is transformed through the vector field.

The distribution at an arbitrary time value $T$ can be trained with maximum likelihood to match data samples, resulting in a generative model of data with the distribution $p(z(T))$. Furthermore, Grathwohl et al. (2019) combines this with a stochastic trace estimator (Hutchinson, 1989) to create an efficient and unbiased estimator of the log probability for data samples $x$.

$$\log p(z(T) = x) = \log p(z(0)) + \mathbb{E}_{v \sim \mathcal{N}(0,1)} \left[ \int_T^0 v^T \left( \frac{\partial f}{\partial z}(t, z(t), \theta) \right) v \, dt \right]. \qquad (6)$$

In Table 2 we show the final test performance and the total number of function evaluations (NFEs) used in the adjoint method over 100 epochs. We see substantially fewer NFEs in experiments on both MNIST and CIFAR-10. Next, we investigate changing model size, by varying the complexity of the vector field $f$, which is a CNN with $d_h$ hidden channels. We find that using the seminorm, the backward NFE does not increase as much as when using the default norm. In particular, we can use roughly same NFEs as the smaller ($d_h = 32$) model to train a larger ($d_h = 128$) model and in doing so achieve a substantial gain in performance ($1.00 \rightarrow 0.95$).

### 3.3 HAMILTONIAN DYNAMICS

Finally we consider the problem of learning Hamiltonian dynamics, using the Symplectic ODE-Net model of Zhong et al. (2020). Under Hamiltonian mechanics, the vector field is parameterised as

$$f(t, z, \theta) = \begin{bmatrix} \frac{\mathrm{d}H}{\mathrm{d}p}(q, p, \theta) \\ g(q, \theta)u - \frac{\mathrm{d}H}{\mathrm{d}q}(q, p, \theta) \\ 0 \end{bmatrix},$$

where $z = (q, p, u)$ is the state decomposed into (generalised) positions $q$, momenta $p$, and control $u$. $H$ is a Hamiltonian and $g$ is an input term, both parameterised as neural networks. The input $g$ offers a way to control the system, which can be understood via energy shaping.

This parameterisation lends itself to being interpretable. $H$ learns to encode the physics of the system in the form of Hamiltonian mechanics, whilst $g$ learns to encode the way in which inputs affect the system. This can then be used to construct controls $u$ driving the system to a desired state. In this context (unlike the other problems we consider here), $z$ is not a hidden state but instead the output of the model. The evolution of $z$ is matched against the observed state at multiple times $z(t_1)$, $z(t_2)$, ..., and trained with $L^2$ loss.

We consider the fully-actuated double pendulum ("acrobot") problem. Training data involves small oscillations under constant forcing. We investigate several quantities of interest.

Table 3: Results for Symplectic ODE-Net on a fully-actuated double pendulum. The number of function evaluations (NFE) used in the adjoint equations—accumulated throughout training—are reported. Test set loss is additionally reported. Mean ± standard deviation over five repeats. The model has 509,108 parameters.

| Default norm | | Seminorm | |
|---|---|---|---|
| Test $L^2$ loss ($10^{-4}$) | Bwd. NFE ($10^4$) | Test $L^2$ loss ($10^{-4}$) | Bwd. NFE ($10^4$) |
| $1.247_{\pm 0.520}$ | $46.45_{\pm 0.01}$ | $2.995_{\pm 2.190}$ | $\mathbf{26.55_{\pm 0.01}}$ |

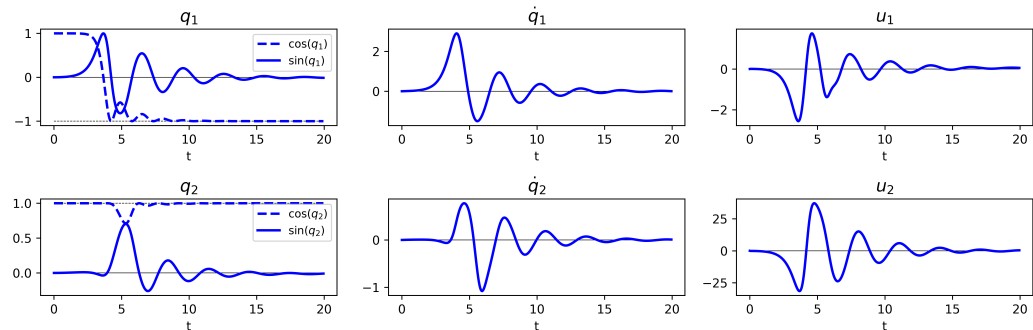

Figure 2: Frames from controlling the fully-actuated double pendulum to the full-upright position.

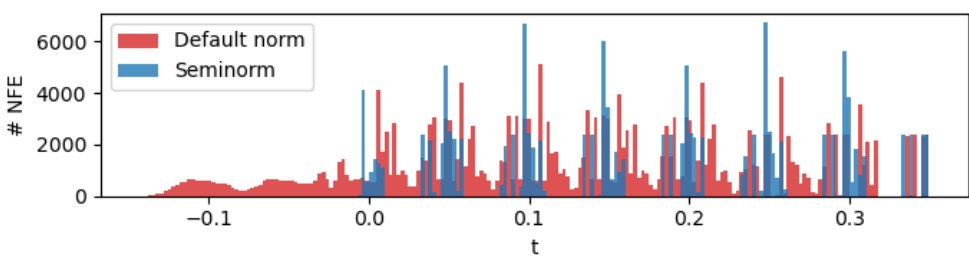

Figure 3: Evolution of the state $z = (q, \dot{q}, u)$ whilst controlling the fully-actuated double pendulum from the full-down to the full-upright position. $q_1$ denotes the angle of the first joint, and $q_2$ denotes the angle of the second joint.

Figure 4: Location of function evaluations for the adjoint equations, during training for the fully-actuated double pendulum problem. All evaluations over five runs are shown.

First we investigate the number of function evaluations. See Table 3. We see that the model successfully learns the dynamics, with very small loss (order $\mathcal{O}(10^{-4})$) in both cases. However, under our proposed change the model is trained using 43% fewer function evaluations on the backward pass.

Next, we verify that the end goal of controlling the system is achievable. See Figures 2 and 3, in which the double pendulum is successfully controlled from the full-down to the full-upright position, using the seminorm-trained model.

After that, we investigate the locations of the function evaluations. See Figure 4. We see that the default norm makes consistently more evaluations for every time $t$. This is interesting as our initial hypothesis was that there would be more spurious rejections for $t$ near the terminal time $T$, where $a_\theta(t)$ is small, as $a_\theta(T) = 0$. In fact we observe consistent improvements for all $t$.

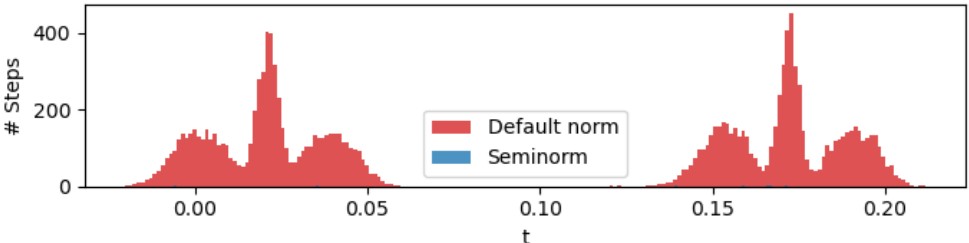

Figure 5: Location of rejected steps during training for the fully-actuated double pendulum problem. All steps over five runs are shown. Rejected steps for the seminorm can just barely be seen.

To conclude, we investigate the locations of the rejected steps. See Figure 5. Note that the Dormand–Prince solver makes 6 function evaluations per step, which is the reason for the difference in scale between Figures 4 and 5. We see that the seminorm produces almost no rejected steps. This accounts for the tight grouping seen in Figure 4, as for each batch the evaluations times are typically at the same locations. Moreover, we observe an interesting structure to the the location of the rejected steps with the default norm. We suspect this is due to the time-varying physics of the problem.

### 3.4 WHERE DO IMPROVEMENTS COME FROM?

**Parameter-state ratio** We hypothesised that the ratio between size of state and number of parameters may explain the differences between the smaller improvements for continuous normalising flows against the substantial improvements for Hamiltonian neural networks and neural CDEs (corresponding to the proportion of channels that can no longer cause rejections). Given a model with $p$ parameters and state $z(t) \in \mathbb{R}^d$, then seminorms reduce the number of rejection-causing channels from $1 + 2d + p$ to just $1 + 2d$.

We plot the (log) ratio $1+2d+p/1+2d$ against the percentage improvement in backward steps. See Figure 6. We see that this hypothesis partly explains things: broadly speaking continuous normalising flows (which have $\mathcal{O}(10^3)$ state variables) see small improvements, whilst the neural CDEs and Hamiltonian neural networks (with $\mathcal{O}(10^1)$ state variables) see larger improvements.

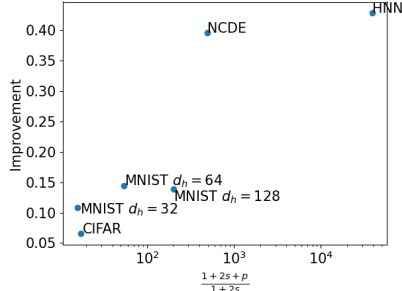

Figure 6: Parameter-state ratio against improvement, as given by the relative decrease in backward NFE. Only one neural CDE experiment (RTOL=$10^{-3}$, ATOL=$10^{-6}$) is shown as the others use different tolerances. CNFs are averaged over their different blocks.

There are still many differences between problems (tolerances, complexity of vector field, and so on); see for example the variation amongst the continuous normalising flows. Thus for any given problem this serves more as a rule-of-thumb.

**Number of accepted/rejected steps** Next we investigate the improvements for each problem, broken down by accepted and rejected steps (over a single backward pass). See Table 4. We begin by noting that the absolute number of accepted and rejected steps nearly always decreased. (In the case of modelling Hamiltonian dynamics, the seminorm actually produced *zero* rejected steps.) However beyond this, the *proportion* of rejected steps nearly always decreased.

This highlights that the benefit of seminorms is two-fold. First, that the number of accepted steps is reduced indicates that use of seminorms means systems are treated as smaller and so easier-to-solve, allowing larger step sizes. Second, the reduced proportion of rejected steps means that the differential equation solve is more efficient, with fewer wasted steps.

## 4 RELATED WORK

Several authors have sought techniques for speeding up training of neural differential equations.

Table 4: Number of rejected and accepted backward steps on one pass over the test set.

| Experiments | Rejected steps ($10^3$) | | Accepted steps ($10^3$) | | Proportion rejected | |
|---|---|---|---|---|---|---|
| | Default | Seminorm | Default | Seminorm | Default | Seminorm |
| NCDE (small) | $\mathbf{0.13}_{\pm\mathbf{0.02}}$ | $0.15_{\pm0.02}$ | $3.45_{\pm0.26}$ | $\mathbf{1.92}_{\pm\mathbf{0.18}}$ | $\mathbf{3.63\%}$ | $7.25\%$ |
| NCDE (medium) | $1.01_{\pm0.18}$ | $\mathbf{0.17}_{\pm\mathbf{0.05}}$ | $8.38_{\pm0.86}$ | $\mathbf{3.25}_{\pm\mathbf{0.43}}$ | $10.76\%$ | $\mathbf{4.97\%}$ |
| NCDE (large) | $10.91_{\pm1.03}$ | $\mathbf{1.16}_{\pm\mathbf{0.13}}$ | $28.86_{\pm2.15}$ | $\mathbf{8.61}_{\pm\mathbf{0.45}}$ | $27.43\%$ | $\mathbf{11.87\%}$ |
| CIFAR-10 | $11.04_{\pm0.29}$ | $\mathbf{9.30}_{\pm\mathbf{1.92}}$ | $23.50_{\pm1.19}$ | $\mathbf{22.82}_{\pm\mathbf{1.34}}$ | $31.96\%$ | $\mathbf{28.95\%}$ |
| MNIST ($d_h = 32$) | $5.54_{\pm0.99}$ | $\mathbf{4.51}_{\pm\mathbf{1.01}}$ | $20.91_{\pm0.90}$ | $\mathbf{19.93}_{\pm\mathbf{0.42}}$ | $20.95\%$ | $\mathbf{18.45\%}$ |
| MNIST ($d_h = 64$) | $7.79_{\pm0.81}$ | $\mathbf{4.29}_{\pm\mathbf{0.69}}$ | $22.95_{\pm1.02}$ | $\mathbf{19.73}_{\pm\mathbf{0.99}}$ | $38.53\%$ | $\mathbf{17.86\%}$ |
| MNIST ($d_h = 128$) | $8.88_{\pm0.21}$ | $\mathbf{7.40}_{\pm\mathbf{1.17}}$ | $26.54_{\pm0.91}$ | $\mathbf{25.01}_{\pm\mathbf{1.17}}$ | $25.07\%$ | $\mathbf{22.83\%}$ |
| HNN | $0.11_{\pm0.00}$ | $\mathbf{0.00}_{\pm\mathbf{0.00}}$ | $0.18_{\pm0.00}$ | $\mathbf{0.16}_{\pm\mathbf{0.00}}$ | $37.93\%$ | $\mathbf{0.00\%}$ |

Ghosh et al. (2020) (much like this work) make a conceptually simple change. They regularise the neural ODE by randomly selecting the terminal integration time. We note that this does not seem applicable to Neural CDEs as well as Neural ODEs.

Finlay et al. (2020) and Kelly et al. (2020) investigate regularising the higher-order derivatives of the model; the idea is that this encourages simpler trajectories that are easier to integrate. However, this improvement must be weighed against the extra cost of computing the regularisation term. Thus whilst Finlay et al. (2020) describe speed improvements for CNFs, Kelly et al. (2020) describe slower training as a result of this extra cost.

Quaglino et al. (2020) describe speeding up training via spectral approximation. Unfortunately, this method is rather involved, and to our knowledge does not have a reference implementation.

When backpropagating through the internal operations of the solver (so not the adjoint method used here), Zhuang et al. (2020) note that backpropagating through rejected steps is unnecessary.

Massaroli et al. (2020b) discuss hypersolvers, which are hybrids of neural networks and numerical differential equation solvers, trained to efficiently solve a desired Neural ODE. However the Neural ODE changes during training, making these most useful after training, to speed up inference.

Dupont et al. (2019) and Massaroli et al. (2020a) note that adding extra dimensions to the Neural ODE improve expressivity and training. Indeed this has arguably now become a standard part of the model; we include this via the linear $\ell_1$ in equation (1).

In the context of Continuous Normalizing Flows, Grathwohl et al. (2019), Chen & Duvenaud (2019), and Onken et al. (2020) have proposed specific architectural choices to reduce training cost.

Chen et al. (2018) mention regularizing the differential equation using weight decay; we include this throughout our experiments.

Independent of modifications to the model or training procedure, Rackauckas & Nie (2017) and Rackauckas et al. (2019) claim speed-ups simply through an improved implementation.

## 5 CONCLUSION

We have introduced a method for reducing the number of function evaluations required to train a neural differential equation, by reducing the number of rejected steps during the adjoint (backward) pass. The method is simple to implement, straightforward to integrate into existing codebases, and offers substantial speed-ups across a variety of applications with no observed downsides.

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

## A  CODE FOR `TORCHDIFFEQ`

Here we discuss in more detail the code used for `torchdiffeq`, that was introduced in Section 2.3.

By default, `torchdiffeq` uses a (slightly odd) mixed $L^\infty$-RMS norm. Given the adjoint state $[a_t, z, a_z, a_\theta]$, then the norm used is

$$\|[a_t, z, a_z, a_\theta]\| = \max\{\|a_t\|_{\mathrm{RMS}}, \|z\|_{\mathrm{RMS}}, \|a_z\|_{\mathrm{RMS}}, \|a_\theta\|_{\mathrm{RMS}}\},$$

where for $x = (x_1, \ldots, x_n)$,

$$\|x\|_{\mathrm{RMS}} = \sqrt{\frac{1}{n} \sum_{i=1}^{n} x_i^2}.$$

Respecting this convention, the code provided reduces this to the seminorm

$$\|[a_t, z, a_z, a_\theta]\| = \max\{\|z\|_{\mathrm{RMS}}, \|a_z\|_{\mathrm{RMS}}\}.$$

Other similar approaches such as pure-RMS over $[z, a_z]$ should be admissible as well.

## B  EXPERIMENTAL DETAILS

### B.1  NEURAL CONTROLLED DIFFERENTIAL EQUATIONS

We use the same setup and hyperparameters as in Kidger et al. (2020). The loss function is cross entropy. The optimiser used was Adam (Kingma & Ba, 2015), with learning rate $1.6 \times 10^{-3}$, batch size of 1024, and 0.01-weighted $L^2$ weight regularisation, trained for 200 epochs. The number of hidden channels (the size of $z$) is 90, and $f$ is parameterised as a feedforward network, of width 40 with 4 hidden layers, ReLU activation, and tanh final activation.

### B.2  CONTINUOUS NORMALISING FLOWS

We follow Grathwohl et al. (2019) and Finlay et al. (2020). The loss function is the negative log likelihood $-\log(p(z(T) = x))$ of equation (6). The optimiser used was Adam, with learning rate $10^{-3}$ and batch size 256, trained for 100 epochs. Relative and absolute tolerance of the solver are both taken to be $10^{-5}$. We used a multi-scale architecture as in Grathwohl et al. (2019) and Finlay et al. (2020), with 4 blocks of CNFs at 3 different scales.

### B.3  SYMPLECTIC ODE-NET

The optimiser used was Adam, with learning rate $10^{-3}$, batch size of 256, and 0.01-weighted $L^2$ weight regularisation. Relative and absolute tolerance of the solver are both taken to be $10^{-4}$. We use the same architecture as in Zhong et al. (2020): this involves parameterising $H$ as a sum of kinetic and potential energy terms. These details of Symplectic ODE-Net are a little involved, and we refer the reader to Zhong et al. (2020) for full details.

