# OpenReview forum: ""Hey, that's not an ODE'": Faster ODE Adjoints with 12 Lines of Code"
_ICLR.cc/2021/Conference — Reject_

### Official Review · AnonReviewer1 · 2020-10-28
**A very clear submission with a very minimal contribution**

**Rating:** 5
**Confidence:** 5

**Review:**

**Summary**
The paper suggests improving the training run-time of neural ODEs when using the adjoint sensitivity method by relaxing the computational steps of $a_{\theta}$ and $a_{t}$.

**Significance**
There are a couple of major issues about the content and the significance of the results presented in this paper:
- While the paper is beautifully organized and easy to follow, I consider the scientific contributions of the work to be limited.
- In the CNF setting, Table 2, I do not see a major improvement in NFE. Why is this the case?
- All neural ODEs have been tested with a single type of ODE solver. I strongly suggest the authors investigate other solvers to see how a model's performance changes when applying the proposed method?
- Deriving a theoretical ground for the reduction in time complexity and its effect on high- and low-frequency time series modeling performance. I suspect the proposed method to speed up the training of neural ODEs would have a significant effect on the performance of time series modeling with high-frequency fluctuations. Could the authors comment on this please?

This paper is a 10 out of 10 as a blog post for the users of the adjoint method for training neural ODEs. I believe that the contribution of this paper is limited to a software hack that would benefit the community best if presented as a blog post.

**Clarity**
An extremely clear submission. I greatly appreciate the authors' effort in preparing such a clear presentation.

**Originality**
The paper is an original contribution.

---

> ### Author Response · Authors · 2020-11-19
> **All points addressed!**
>
> Thank you for your review. Apologies for the delay in responding -- we were performing follow-up experiments (discussed below) in response to these reviews.
>
> The reviewer's primary concern appears to be that the proposed change is small. However, we quite firmly believe that this is a good thing, not a bad thing. The size of a paper's contribution should be proportional to the benefits gained -- in this case, a median 40% improvement, with no observed downsides! -- and not to the complexity of the technique proposed.
>
> We are very pleased that they describe the paper as "beautifully organised" and "extremely clear". (If only every paper was described that way.)
>
> Regarding the other questions:
>
> - We have now introduced an additional set of experiments investigating the source of the improvement, such as the smaller improvement for CNFs. This is given in the new Section 3.4.
>
>   We see that the difference between problems is roughly attributable to the ratio between the number of parameters and the size of the state. This corresponds to the the reduction in rejection-causing channels on the adjoint pass. For example, CNF models have large state but (relatively) few parameters, whilst the other experiments have small state but many parameters.
>
>   It was for this reason that we performed ablation studies with CNF models that are larger than the standard used in previous works. This not only led to a larger gap in NFE between the standard norm and our seminorm, but also to an improvement from the original FFJORD paper's 3.35->3.34 and 0.99->0.95 for CIFAR10 and MNIST respectively. These numbers can be contrasted with the explicit regularisation of Finlay et al. 2020, which are 3.38 and 0.97. Their regularisation performs worse than vanilla CNF on CIFAR10 whilst our method is simpler to implement, does not introduce additional computational cost, and does not hinder performance at all.
>
> - Comparisons to other ODE solvers would be a good fit; indeed we expect that our contribution should readily be applicable to any solver scheme that can solve z reliably. However given the ubiquity of Dormand--Prince amongst adaptive solvers, this was simply judged to be less important than the other experiments already present in the paper, such as validating the technique across a wide range of domains.
>
> - On deriving a theoretical description: we are not aware of a theory for analysing the expected number of steps of an adaptive step-size ODE solver; every comparison we are aware of is empirical. If the reviewer is familiar with a literature on this topic we would be grateful to know more. (But otherwise of course we cannot update the paper to discuss what does not exist.)
>
> **Summary**
> Thankyou again for your review. We hope that this rebuttal, and the additional experiments, have offset the reviewer's concerns. We would welcome further discussion towards improving their score.

---

### Official Review · AnonReviewer3 · 2020-10-28
**Small and easy proposed change, but more experiments required to check consistency of conclusions**

**Rating:** 5
**Confidence:** 5

**Review:**

Update

I have revised my rating based on the updates. The paper has good theoretical insights, however I agree with the other reviewers that more completeness is required to make the proposal stronger.

--------------------------

Summary
The paper recognizes that the error calculations in Neural ODEs involve some terms that don’t need to be accounted for. It then shows that by ignoring those terms, the number of function evaluations reduces while approximately maintaining accuracy. The paper demonstrates this in 3 tasks - speech classification using Neural CDEs, generative modeling using FFJORD, and Hamiltonian dynamics using Symplectic ODE-Net.

Strengths
The paper is well written, the motivation is clearly explained. There is a diverse set of tasks that are tackled, which is great to check for consistency of the conclusions drawn. A code snippet is provided for one implementation to show how simple the proposed change is to incorporate into existing code. Where possible, helpful graphs have been provided to help visualize the conclusions of the respective experiments. The paper also mentions other works and how this work is not closely related to them.

Comments
Different aspects of introducing the paper’s contribution are explored only in one task, and not all. For example, the graph of the Backward NFE vs Epoch is showcased for the 1st task of speech classification, but not the rest of the tasks. It would be important to understand if the conclusion that the semi-norm has better NFE throughout training is consistent in all the tasks explored. For example, the proposed change does not seem to be very effective in the task of generative modelling in terms of BPD, hence it would be important to understand how the proposed change affects training in that task.

This also holds for locations of rejection steps, as it was mentioned that although intuitively the spurious steps closer to T=0 would be rejected but empirically it holds true for all t, but this aspect is only explored in the 3rd task. It would be important to understand how consistent this is across tasks.

Overall, the paper seems to introduce a minor change that only works on simpler tasks but does not affect higher-dimensional tasks such as generative modeling very effectively. This is probably because the underlying dynamics for the generative modeling tasks are necessarily complex. Works such as Finlay et al (2020) and Ghosh et al (2020) (uncited in the paper!) propose to reduce the complexity of the underlying dynamics. It is pertinent to see how this paper’s proposed change affects the training of those models. Hopefully, the simpler dynamics should implicitly lend to smaller NFEs, which the proposed change can take advantage of and show faster results on.

It is important to note that terms such as “standard” may not apply to Neural ODEs, since they are hardly 2 years old and so are quite nascent and yet to be widely adopted. The fact that some papers that work on Neural ODEs are not as cited or implemented in software packages as others does not mean they are not “standard” (or any such related term), and vice-versa.

The “12 lines of code” contribution also only works for the “standard” implementation. This is not a weakness per se, but software-related changes are highly subject to the respective software being used. Something that’s 12 lines in one particular code repository implemented in PyTorch may be a lot more lines in a different implementation or in a different language such as Julia or Jax.

Finlay et al. (2020) := How to Train Your Neural ODE; ICML 2020
Ghosh et al. (2020) := STEER: Simple Temporal Regularization For Neural ODEs ; NeurIPS 2020; https://arxiv.org/pdf/2006.10711.pdf

---

> ### Author Response · Authors · 2020-11-19
> **Response**
>
> Thank you for your review. Apologies for the delay in responding -- we were performing follow-up experiments (discussed below) in response to these reviews.
>
> The reviewer notes that different topics are only explored with different tasks. This was a deliberate choice -- repeating every graph for every task just did not seem like an interesting presentation of the material (or an efficient use of space). But for completeness: the behaviour was indeed consistent across every task.
>
> The reviewer's other main criticism seems to be that the topic of the paper is a "minor change that only works on simpler tasks". First, we quite firmly that a paper's contribution should be proportional to its efficacy -- in our case, median 40% improvement, with no observed downsides -- not to the complexity of its ideas. (Indeed Ghosh et al. 2020, highlighted by the reviewer, is an improvement that is also conceptually very simple.) Second, we do not regard any of the experiments as a "simpler task". Every problem was selected as being one of important practical relevance. It simply isn't the case that every interesting or difficult neural ODE is a CNF.
>
> In response to the more minor remarks:
>
> - We agree that the "12 line of code" refers to a particular implementation. It was included -- a little hyperbolically -- to emphasise the simplicity of the change.
> - Ghosh et al (2020) was unpublished when this paper was submitted for review. We have now included a reference.
> - The term "standard" has been replaced with "reference".
>
> On a more positive note, we are happy to see that the reviewer identifies this paper as clearly written, well motivated, and that the diverse set of tasks is "great for consistency of the conclusions drawn".
>
> Additionally, we have now updated the paper with a whole new set of experiments (the new Section 3.4), in which we investigate where the benefits come from, and why some tasks produce greater improvements than others; in particular with respect to the ratio between the number of parameters and the size of thes state.
>
> **Summary**
> We hope that this ameliorates the reviewer's concerns, and we would welcome any further discussion towards improving their score.

---

### Official Review · AnonReviewer4 · 2020-10-28
**A practical trick to improve the efficiency of the adjoint method in neural ODEs.**

**Rating:** 4
**Confidence:** 3

**Review:**

Summary: The paper proposes a modification for the adjoint method, such that to improve the training efficiency of neural ODEs. The proposed idea is that the solution of some terms in the adjoint method can be less accurate, because these are not ODEs but simple integrals, and hence, the error does not propagate. Thus, the solver can utilize bigger steps, and in total to perform less steps. In the experiments the efficiency is demonstrated under different scenarios where neural ODEs are used.

Comments: At first, I think that the proposed idea is quite interesting. Even if I am not an expert in the field, the theoretical argumentation seems to be rather reasonable. Also, the proposed trick seems to improve in practice the efficiency of the method.

However, I think that the writing of the paper can be improved. In my opinion, at the current stage the paper looks like a collection of (technical) notes, but probably not as a proper conference paper. Since the proposed idea is a rather simple trick, I believe that is necessary a better description of the technical background, for example the adjoint method and the whole neural ODEs procedure. In my opinion, the paper is written in way which assumes that the reader is very familiar with the topic and its corresponding details. Also, I find the coherence of the story a bit lacking. Therefore, I believe, that improving the paper along these line is essential.

===== After rebuttal =====

As I have mentioned in my review, I am not an expert in the field of neural ODEs. Regarding the technical contribution I think is quite interesting, but probably rather small as far as I understand from the rest of the reviews. In my opinion, the current paper is very much focused to audience which is very familiar with the topic, and its impact on a non-expert probably limited. Therefore, I tend to keep my score and vote for rejection, because I feel that the submission has to become more accessible to general audience.

---

> ### Author Response · Authors · 2020-11-19
> **Response**
>
> Thank you for your review. Apologies for the delay in responding -- we were performing follow-up experiments in response to these reviews. (Experiments discussed in our responses to the other reviewers, who requested this.)
>
> The reviewer's only criticism seems to be the presentation of the material. This is inconsistent with reviewers 1, 3 and 4, who all praise the presentation.
>
> Certainly we assume some familiarity with the topic of neural ODEs -- as this is a paper about neural ODEs.
>
> If the reviewer can offer more specific feedback on the presentation then we would welcome the opportunity, but otherwise we would ask them to reconsider their score.

---

### Official Review · AnonReviewer2 · 2020-10-28
**A simple and straightforward trick to reduce NFEs during the backward pass. There are some questions on experimental comparisons.**

**Rating:** 5
**Confidence:** 4

**Review:**

Summarizing the paper claims
------------------------------------------
The paper addresses the problem of reducing the number of function evaluations (NFEs) during neural ODE training with adaptive solver and the adjoint method. Namely, the authors claim that for the variety of applications,  NFEs at backward pass can be reduced if the automated step selection will be based only on the part of the information of the adjoint state.

Strong points
-------------------
The paper is clearly written. The performance of the proposed method is evaluated for a wide variety of tasks (time series, generative modeling, physical control). Hyperparameter setups are provided to reproduce the experiments.

Weak points
-----------------
The paper doesn't provide a comparison with the existing techniques for speeding up NFEs (e.g., Finlay et al. (2020)).
The paper claims that the proposed approach is complementary to the techniques mentioned in the related paper. However, I haven't found supportive experiments for this assumption.  If I understood correctly,  the authors use weight decay everywhere in their experiments. Therefore, I wonder if the proposed method shows similar behavior when weight decay is omitted.

Recommendation (accept or reject)
-------------------------------------------------
For this moment, I'd recommend rejecting the paper due to the mentioned weak points; however, I consider updating the decision if weak points and questions will be closed during the rebuttal period.

Update: Thanks to the authors for the work done during the rebuttal period. The proposed method seems working but it's still not very clear when it is beneficial to use it. The authors provided some intuition about the cases when the method is effective, however, I think it is necessary to more thoroughly explain the applicability of the method before sharing it with the community at the conference, that is why I don't change my score.

Questions
--------------
I'd like to understand why for different applications relative influence of this approach is quite different? For example, for Continuous Normalising Flows (Table 2), the gain is not so significant as for Neural CDEs (Table 1). What are the crucial properties of the task to benefit a lot from the proposed technique? Are there any limitations?

Describing Symplectic ODE-Net results in Table 3, the authors claim that the loss is almost the same for default norm/seminorm training. However, the reported mean loss for seminorm is twice bigger, and the reported standard deviation is four times bigger than corresponding values for the default norm. Hence, that would be nice to see NFEs comparison when loss statistics are closer, because for now, that is not  clear whether there is a real improvement due to the proposed method or this is a  trade-off between loss statistics and the number of NFEs

---

> ### Author Response · Authors · 2020-11-19
> **Questions addressed, and additional experiments introduced.**
>
> Thank you for your review. Apologies for the delay in responding -- we were performing follow-up experiments (discussed below) in response to these reviews.
>
> We agree that the claim that techniques may be complementary is unsubstantiated and have removed it from the paper.
>
> We agree that further experiments (comparisons to other techniques, varying the amount of weight decay) could be performed. We believed that this was unnecessary as the existing experiments already demonstrate that our technique already has a large (~40%) direct benefit, with no observed downsides.
>
> - However, we have now introduced additional experiments in response to queries by both this reviewer and Reviewer 1, seeking to better explain the source of the improvements when using seminorms. This is given in the new Section 3.4.
>
>   In particular, we see that the difference between problems is roughly attributable to the ratio between the number of parameters and the size of the state. This corresponds to the the reduction in rejection-causing channels on the adjoint pass. For example, CNF models have large state but (relatively) few parameters, whilst the other experiments have small state but many parameters.
>
>   It was for this reason that we performed ablation studies with CNF models that are larger than the standard used in previous works. This not only led to a larger gap in NFE between the standard norm and our seminorm, but also to an improvement from the original FFJORD paper's 3.35->3.34 and 0.99->0.95 for CIFAR10 and MNIST respectively. These numbers can be contrasted with the explicit regularisation of Finlay et al. 2020, which are 3.38 and 0.97. Their regularisation performs worse than vanilla CNF on CIFAR10 whilst our method is simpler to implement, does not introduce additional computational cost, and does not hinder performance at all.
>
>   The paper has been updated to reflect these points.
>
> - The loss for Symplectic ODE-Net is almost the same for default vs seminorm. They differ by only $O(10^{-4})$, and both models train to match the data essentially perfectly. For very small quantities, multiplicative descriptions ("twice as large") are typically less appropriate.
>
> **Summary**
> We are pleased that the reviewer identifies the paper as clearly written, and that they are open to updating their score. We would be very happy to have further dialogue.

---

> > ### Comment · AnonReviewer2 · 2020-11-21
> > **Questions on the recently provided arguments**
> >
> > Thank you for addressing the questions. There are some questions about the newly added parts.
> >
> > 1. As it follows from the Appendix (section B.2), you use multi-scale architectures for experiments with CNFs. Therefore, if I understand correctly, one architecture contains ODE blocks with different state sizes. How for such architectures you compute the ratio (1 + 2s + p / 1 + 2s), which is depicted in Figure 6?
> >
> > 2.  Which NCDE architecture is depicted in Figure 6, small/medium or large? I suggest adding all three to the figure.
> >
> > 3. Another concern follows from the comparison of Table 2 and Table 4. Do I understand correctly that for every single backward pass, (the number of rejected steps plus the number of accepted steps) multiplied by (NFEs per step) should be equal to the (number of backward NFEs)? If so, for a fixed dataset and architecture, the value from Table 2 (column "Seminorm/Bwd.NFE") should be approximately equal to the sum of two values from Table 4 (columns "Rejected steps/Seminorm", "Accepted steps/Seminorm") multiplied by 6 (if the case of dopri4/5). However, that doesn't hold. Consider, for example, MNIST (d_h=32), its "Seminorm/Bwd.NFE)" equals 37.03 +/- 0.96, while "Rejected steps/Seminorm" and "Accepted steps/Seminorm" are 4.51 +/- 1.01 and 19.93 +/- 0.42.
> > Could you please, explain why such an inconsistency in numbers occurs?
> >
> > 4. For CNF models, I would recommend plotting (at least in Appendix) Bwd. NFE vs. epoch, Rejected steps vs. epoch, and Accepted steps vs. epoch. That will make arguments in section 3.4 clearer.
> >
> > 5. Moreover, I think providing for different tasks loss vs. real-time plots will further improve the paper, especially from the application perspective.

---

> > > ### Author Response · Authors · 2020-11-21
> > > **Questions addressed**
> > >
> > > 1. Your understanding is correct. We average the results over the blocks to obtain the plot shown; this is now stated explicitly in the caption.
> > >
> > > 2. The RTOL,ATOL=1e-3,1e-6 version is used. The others are deliberately omitted as using very different tolerances to the other problems considered. We have included this in the caption.
> > >
> > > 3. Table 2 states the accumulated NFE over all epochs of training, while Table 4 states the accepted/rejected steps over a pass on the test set. As such these numbers show different properties.
> > >
> > > 4./5. We will keep these suggestions in mind. However, the paper already states much of this information in other forms -- for example the total NFE over training is already reported in Table 2, which reflects the total training cost. This is a reproducible and interpretable metric that can be compared across papers. Respectfully, we feel that there are already more than enough plots to justify the efficacy of our approach.

---

### Public Comment · ~Juntang_Zhuang1 · 2020-11-13
**Missing reference to a related work**

Hi, thanks for the nice paper. We noticed that your proposed method is aimed to speedup the back-propagation in continuous case, and our ICML paper proposed the ACA method, which is a new numerical method for back-propagation in Neural ODE. ACA reduces the training time by almost half in Fig7(b) of [1], and achieves better accuracy than the adjoint method. We would appreciated it if you could briefly discuss.

[1] Zhuang, Juntang, et al. "Adaptive Checkpoint Adjoint Method for Gradient Estimation in Neural ODE." arXiv preprint arXiv:2006.02493 (2020).

---

> ### Author Response · Authors · 2020-11-19
> **Response**
>
> Thanks for your interest. We've added a brief reference.

---

### Decision · Program_Chairs · 2021-01-07
**Final Decision**

**Decision:**

Reject

**Comment:**

The paper focuses on NeuralODE and shows that for the implementation popular among ML community, one of the equations is not an ODE and can be replaced by an integral. This is implemented using "seminorm" (just assigning zero weight to the last equation).

Pros:
- Well written
- Useful to replace the "standard" implementation
-  Consistent benchmarking

Cons:
- Contribution is too limited
- Used in several "prior" codes without explicit ICLR submission.
- (My personal) The title is not good: more on the "hype side" of the story, rather than progressing the field. I don't think we need to put every single minor fact into a ICLR submission.  For example, one can just compute the integral as an alternative by any suitable quadrature rule. That would add 10-15 function evaluations at most, since most of the functions in NeuralODEs are quite smooth.